# Interpreting Chest X-rays via CNNs that Exploit Hierarchical Disease Dependencies and Uncertainty Labels

**Hieu H. Pham**[1]                                                       v.hieuph4@vintech.net.vn

**Tung T. Le**[1]                                                  lethanhtung.tung06@gmail.com

**Dat T. Ngo**[1]                                                      v.datnt41@vintech.net.vn

**Dat Q. Tran**[1]                                                     v.dattq13@vintech.net.vn

**Ha Q. Nguyen**[1]                                                    v.hanq3@vintech.net.vn

[1] *Medical Imaging Group, Vingroup Big Data Institute (VinBDI), Hanoi, Vietnam*

## 1. Introduction

The chest X-rays (CXRs) is one of the views most commonly ordered by radiologists (NHS), which is critical for diagnosis of many different thoracic diseases. Accurately detecting the presence of multiple diseases from CXRs is still a challenging task. We present a multi-label classification framework based on deep convolutional neural networks (CNNs) for diagnosing the presence of 14 common thoracic diseases and observations. Specifically, we trained a strong set of CNNs that exploit dependencies among abnormality labels and used the label smoothing regularization (LSR) for a better handling of uncertain samples. Our deep networks were trained on over 200,000 CXRs of the recently released CheXpert dataset (Irvin and *al.*, 2019) and the final model, which was an ensemble of the best performing networks, achieved a mean area under the curve (AUC) of 0.940 in predicting 5 selected pathologies from the validation set. To the best of our knowledge, this is the highest AUC score yet reported to date. More importantly, the proposed method was also evaluated on an independent test set of the CheXpert competition, containing 500 CXR studies annotated by a panel of 5 experienced radiologists. The reported performance was on average better than 2.6 out of 3 other individual radiologists with a mean AUC of 0.930, which had led to the current state-of-the-art performance on the CheXpert test set.

## 2. Proposed approach

### 2.1. Dataset and settings

Our focus in this paper is to develop and evaluate a deep learning-based approach that could learn from hundreds of thousands of CXR images and make accurate diagnoses of 14 common thoracic diseases and observations (Rajpurkar et al., 2017, 2018). The CheXpert dataset was used to train and validate the proposed method. It contains 224,316 scans of 65,240 patients, annotated for the presence of 14 common chest CRX observations. Each observation can be assigned to either *positive* (1), *negative* (0), or *uncertain* (-1). The whole dataset is divided into a training set of 223,414 studies, a validation set of 200 studies, and a hidden test set of 500 studies. For the validation set, each study is annotated by 3 board-certified radiologists and the majority vote of these annotations serves as the ground-truth. Meanwhile, each study in the hidden test set is labeled by the consensus of 5 board-certified radiologists. The main task on the CheXpert is to build a classifier that takes as input a CXR image and outputs the probability of each of the 14 labels. The effectiveness is measured by the AUC metric over 5 selected diseases (*i.e.* *Atelectasis*, *Cardiomegaly*, *Consolidation*, *Edema*, and *Pleural Effusion*) as well as by a reader study.

## 2.2. Exploiting disease dependencies and dealing with uncertainty labels

In CXRs, diagnoses are often conditioned upon their parent labels and organized into hierarchies (Van Eeden et al., 2012). Most existing CXR classification approaches, however, treat each label in an independent manner. This paper proposes to build a deep learning system that is able to take the label structure into account and learn label dependencies. To this end, we train CNN-based classifiers on conditional data with all parent-level labels being positive and then finetune them with the whole dataset. We adapt the idea of conditional learning (Chen et al., 2019) to the lung disease hierarchy of the CheXpert dataset (Irvin and al., 2019). First, a CNN is pretrained on a partial training set containing all positive parent labels to get more accurate predictions of child labels (Figure 1a). Next, transfer learning will be exploited. We freeze all the layers of the pretrained network except the last fully connected layer and then retrain it on the full dataset. This stage aims at improving the capacity of the network in predicting parent-level labels. During the inference phase,

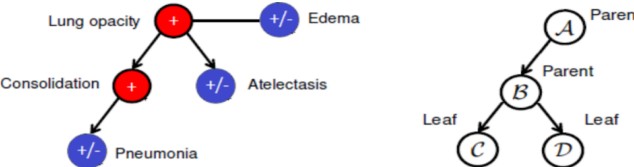

Figure 1: (a) A CNN is trained on a training set where all parent labels (red nodes) are positive, to classify leaf labels (blue nodes), which could be either positive or negative . (b) An example of a tree of 4 diseases: $\mathcal{A}$, $\mathcal{B}$, $\mathcal{C}$, and $\mathcal{D}$ that depend on each other.

all the labels should be unconditionally predicted. Thus, as a simple application of the Bayes rule, the unconditional probability of each label being positive should be computed by multiplying all conditional probabilities produced by the trained CNN along the path from the root node to the current label. For example, let $\mathcal{C}$ and $\mathcal{D}$ be disease labels at the leaf nodes of a tree $\mathcal{T}$ (Figure 1b), which also parent labels $\mathcal{A}$ and $\mathcal{B}$. Suppose the tuple of conditional predictions $(p(\mathcal{A}), p(\mathcal{B}|\mathcal{A}), p(\mathcal{C}|\mathcal{B}), p(\mathcal{D}|\mathcal{B}))$ are already provided by the trained network. Then, the unconditional predictions for the presence of $\mathcal{C}$ and $\mathcal{D}$ will be computed as $p(\mathcal{C}) = p(\mathcal{A})p(B|\mathcal{A})p(\mathcal{C}|\mathcal{B})$ and $p(\mathcal{D}) = p(\mathcal{A})p(B|\mathcal{A})p(\mathcal{D}|\mathcal{B})$.

Another challenging issue in the classification of CXRs is that the training dataset have many CXR images with uncertainty labels. Several approaches have been proposed (Irvin and al., 2019) to deal with this problem. E.g, they can be all *ignored* (U-Ignore), all mapped to *positive* (U-Ones), or all mapped to *negative* (U-Zeros). Unlike previous works, we propose to apply the LSR (Muller et al., 2019) for a better handling of uncertainty samples. Specifically, the U-ones approach is softened by mapping each uncertainty label $(-1)$ to a random number close to 1. The proposed U-ones+LSR approach now maps the original label $y_k^{(i)}$ to

$$\bar{y}_k^{(i)} = \begin{cases} u, & \text{if } y_k^{(i)} = -1 \\ y_k^{(i)}, & \text{otherwise,} \end{cases} \tag{1}$$

where $u \sim U(a_1, b_1)$ is a uniformly distributed random variable between $a_1$ and $b_1$–the hyper-parameters of this approach. Similarly, we propose the U-zeros+LSR approach that softens the U-zeros by setting each uncertainty label to a random number $u \sim U(a_0, b_0)$ that is closed to 0.

### 2.3. Deep learning model and training procedure

We trained and evaluated DenseNet-121 (Huang et al., 2017) as a baseline model on the Chexpert dataset to verify the impact of the proposed conditional training procedure and LSR. Then, a strong set of different state-of-the-art CNNs have been experimented including: DenseNet-121, DenseNet-169, DenseNet-201 (Huang et al., 2017), Inception-ResNet-v2 (Szegedy et al., 2017), Xception (Chollet, 2017), and NASNetLarge (Zoph et al., 2018). In the training stage, all images were fed into the network with a standard size. Before that, a template matching algorithm (Brunelli, 2009) was used to search and find the location of lungs on the original images, which helps to remove the irrelevant noisy areas such as texts or the existence of irregular borders. The final fully-connected layer is a 14-dimensional dense layer, followed by sigmoid activations that were applied to each of the outputs to obtain the predicted probabilities of the presence of the 14 pathology classes. We used Adam optimizer (Kingma and Ba, 2015) with default parameters $\beta_1 = 0.9$, $\beta_2 = 0.999$ and a batch size of 32 to find the optimal weights. The learning rate was initially set to $1e - 4$ and then reduced by a factor of 10 after each epoch during the training phase. Our network was initialized with the pretrained model on ImageNet (Krizhevsky et al., 2012) and then trained for 50,000 iterations. The ensemble model was simply obtained by averaging the outputs of all trained networks.

## 3. Experiments and results

Our extensive ablation studies show that both the proposed conditional training and LSR helped boost the model performance. The baseline model trained with the `U-Ones+CT+LSR` approach obtained an AUC of 0.894 on the validation set. This was a 4% improvement compared to the baseline trained with the `U-Ones` approach that obtained a mean AUC of 0.860. Our final model, which was an ensemble of six single models, achieved a mean AUC of 0.940 – a score that outperforms all previous state-of-the-art results (Irvin and *al.*, 2019; Allaouzi and Ahmed, 2019) by a big margin. To compare our model with human expert-level performance, we evaluated the ensemble model on the hidden test set and performed ROC analysis. The ROCs produced by the prediction model and the three radiologists' operating points were both plotted. For each disease, whether the model is superior to radiologists' performances was determined by counting the number of radiologists' operating points lying below the ROC. The result shows that our deep learning model, when being averaged over the 5 diseases, outperformed 2.6 out of 3 radiologists with a mean AUC of 0.930. This is the best performance on the CheXpert leaderboard at the time of writing this paper.

## 4. Conclusion

We presented in this paper a deep learning-based approach for building a high-precision computer-aided diagnosis system for common thoracic diseases classification from CXRs. In particular, we introduced a new training procedure in which dependencies among diseases and uncertainty labels are effectively exploited and integrated in training advanced CNNs. Extensive experiments demonstrated that the proposed method outperforms the previous state-of-the-art by a large margin on the CheXpert dataset. More importantly, our deep learning algorithm exhibited a performance on par with specialists in an independent test.

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
