# OpenReview forum: "Interpreting Chest X-rays via CNNs that Exploit Hierarchical Disease Dependencies and Uncertainty Labels"
_MIDL.io/2020/Conference — MIDL 2020_

### Official Review · AnonReviewer2 · 2020-03-05
**An interesting CNN strategy to robustly predict the presence of 14 common thoracic diseases and observations**

**Rating:** 3
**Confidence:** 2

**Review:**

This paper presents a multi-label classification framework based on deep convolutional neural networks (CNNs) for diagnosing the presence of 14 common thoracic diseases and observations in X-rays images. The novelty of the proposed framework is to take the label structure into account and to learn label dependencies, based on the idea of conditional learning in (Chen et al., 2019) and the lung disease hierarchy of the CheXpert dataset (Irvin and al., 2019). The method is then shown to significantly outperform the state-of-the-art methods of (Irvin and al., 2019; Allaouzi and Ahmed, 2019).

The paper reads well and the methodology seems to be interesting. I only regret the fact that this is a short paper, and there is therefore not enough space for a more formal description and discussion of the methodology.

---

### Official Review · AnonReviewer1 · 2020-03-12
**The claimed results are not well presented or proven**

**Rating:** 1
**Confidence:** 5

**Review:**

The authors present a work that classifies chest x-ray images with 14 different labels and uses hierarchical labelling and label regularization in an attempt to improve results.

A leading performance on the public chexpert challenge is claimed, but while the authors may have created a nice model the claims they make in this paper are not well proven or explained.

The method for using hierarchical labelling appears to follow a previously published scheme (cited) except with a different hierarchy (no details of the new hierarchy are provided).  The method for label regularization is also previously published (and cited), therefore there is not methodological novelty in the paper.

The authors apply their methods to the chexpert public dataset.   From section 2.3 it is not clear to me precisely what experiments were carried out - were all of these models trained with/without the hierarchical labelling and also with/without the label regularization?  That is not described at all.

Section 3 claims that extensive ablation studies were carried out, however there is not a single table or figure to illustrate the results of these.  The text provides a few AUC values but the precise gain from the hierarchical labelling and from the label regularization is unclear.   What is meant by "U-ones+CT+LSR" - this is mentioned in results but not explained.

The paper has no abstract.

---

### Official Review · AnonReviewer3 · 2020-03-13
**Most part of the paper is clear, but the experiment set-up may not theoretically support the proposed algorithm.**

**Rating:** 3
**Confidence:** 3

**Review:**

This short paper proposes exploit dependencies among abnormality labels and used the label smoothing regularization for a better handling of uncertain samples.
Pros:
1. The proposed model gains 4% improvement in AUC from the label smoothing regularization compared with pure U-Ones.
2. The proposed work achieves the highest AUC for 5 selected pathologies
3. The proposed work is on average better than 2.6 out of 3 other individual radiologists.
Cons:
1. All 14 labels are trained, but the model only has 14 outputs. Does that mean "parent labels" in the paper are labels included in the dataset? If so, is it guaranteed that parent is positive when at least one child is positive? This is the essential assumption in the adapted model (Chen et al. 2019).
2. Terms not consistent: "we propose the U-zeros+LSR approach" at the end of Section 2.2. But U-Ones+LSR is evaluated in ablation study.
3. Lack ablation study with the model ignoring all uncertain cases. (defined as U-Ignore in the paper)

---

### Official Review · AnonReviewer4 · 2020-03-19
**Dual submission?**

**Rating:** 1
**Confidence:** 3

**Review:**

I was searching for relevant work but found an arXiv paper that is very similar and being reviewed under Neurocomputing Journal: https://arxiv.org/abs/1911.06475

This paper is poorly written and not well organized. It is unclear to me how the method works and the results section is also not informative.

---

> ### Comment · Program_Chairs · 2020-04-06
> **Journal submission no problem**
>
> Please note that MIDL 2020 allows short papers discussing recently published or submitted journal contributions - does this change your assessment?

---

> ### Comment · Area_Chair1 · 2020-04-07
> **re-evaluate the scientific value of the paper**
>
> Dear Reviewer,
>
> In this case, would you like to re-evaluate this paper regarding its scientific value? And let us know your final decision asap, thank you.

---

### Meta-Review · Area_Chair1 · 2020-04-04
**MetaReview of Paper23 by AreaChair1**

**Rating:** 2

**Metareview:**

First of all, this paper does not follow the MIDL template, with missing the Abstract section. Major concerns from the reviewers lie in the unclear presentation of results and large overlapping with an arXiv paper. Nevertheless, I think that taking into account the structural dependencies of labels is interesting.

**Paper Type:**

methodological development

---

### Decision · Program_Chairs · 2020-04-11

**Decision:**

Accept

**Comment:**

Taking all information into account and noting a misunderstanding on the reviewer's part on MIDL dual submisison policy it was determined that the paper was accepted based on its merit.